# The Children's Hospitals in Africa Mapping Project (CHAMP) survey: Facilities, equipment, supplies, infrastructure, and capacity to respond to emergencies

Vinayak Bhardwaj[1]⊙, Lawrence R. Stanberry[2]⊙, Philip LaRussa[2]⊙, Wilmot James[3]⊙, Maitry Mahida[2]⊙, Aimable Kanyamuhunga[4], Atnafu Mekonnen Tekleab[5], Augustine Omoigberale[6], Crispen Ngwenya[7], David Musorewegomo[8], Dipesalema Joel[9], Ezekiel Mupere[10], Fidelis Ewenitie Eki-Udoko[6], Hannah Bousquet[2], Heloise Buys[11], Hilda Angela Mujuru[8], Ike Oluwa Lagunju[12], Irene Marete[13], Jethro Zawolo[14], Jonathan Kaunda Mwansa[15], Joseph Tawanda Chava[8], Maima Kawah Baysah[14], Mildred Anyango Mudany[16], Nancy Biyeah Yang Ngum[17], Nellie V. T. Bell[18], One Bayani[9], Pauline Samia[19], Ruth Nduati[20], Sam Miti[21], Schyler Zane Grodman[22], Thembisile Dintle Mosalakatane[9], Workeabeba Abebe[23], Ashraf Coovadia[24]⊙ *

1 School of Public Health, University of the Witwatersrand, Johannesburg, South Africa, 2 Department of Pediatrics, Columbia University Irving Medical Center, New York, New York, United States of America, 3 School of Public Health, Pandemic Center, Brown University, Providence, Rhode Island, United States of America, 4 Pediatrics & Child Health Department, College of Medicine and Health Sciences, University of Rwanda, Kigali, Rwanda, 5 Department of Pediatrics and Child Health, St Paul's Hospital Millennium Medical College, Addis Ababa, Ethiopia, 6 Department of Child Health, University of Benin Teaching Hospital, Benin City, Edo State, Nigeria, 7 Department of Paediatrics, Faculty of Medicine, Midlands State University, Gweru, Zimbabwe, 8 Department of Child, Adolescent and Women Health, Faculty of Medicine and Health Sciences, University of Zimbabwe, Harare, Zimbabwe, 9 Department of Paediatrics and Adolescent Health, Faculty of Medicine, University of Botswana, Gaborone, Botswana, 10 Department of Paediatrics and Child Health, School of Medicine College of Health Sciences, Makerere University, Kampala, Uganda, 11 Department of Paediatrics & Child Health, University of Cape Town, Cape Town, South Africa, 12 Department of Paediatrics, College of Medicine, University of Ibadan, Ibadan, Nigeria, 13 Department of Child Health and Paediatrics, School of Medicine, Moi University, Cheptiret, Kenya, 14 A.M. Dogliotti College of Health Sciences, University of Liberia, Congo Town, Monrovia, Liberia, 15 USAID Stop GBV Now Project, Zambia Center for Communication Programmes (Kwatu), Ministry of Health, Ndola, Zambia, 16 Independent Researcher, Help Reach Africa, Nairobi, Kenya, 17 Africa Medicine Regulatory Harmonisation Programme (AMRH), African Union Development Agency, AUDA-NEPAD, Pretoria, South Africa, 18 Department of Paediatrics, Ola During Children's Hospital, University of Sierra Leone Teaching Hospitals Complex, Freetown, Sierra Leone, 19 Department of Paediatrics and Child Health, Medical College, Aga Khan University, Nairobi, Kenya, 20 School of Medicine, College of Health Sciences, University of Nairobi, Nairobi, Kenya, 21 Department of Clinical Sciences, National Health Research and Training Institute (formerly known as TDRC), Ndola, Zambia, 22 Division of Disaster Medicine, Beth Israel Deaconess Medical Center, Boston, Massachusetts, United States of America, 23 Department of Paediatrics and Child Health, College of Health Sciences, Addis Ababa University, Addis Ababa, Ethiopia, 24 Department of Paediatrics and Child Health, School of Clinical Medicine, Faculty of Health Sciences, University of the Witwatersrand, Johannesburg, South Africa

⊙ These authors contributed equally to this work.
* ashraf.coovadia@wits.ac.za

## Abstract

The Children's Hospitals in Africa Mapping Project (CHAMP) survey was developed and implemented to assess the capabilities of some of the best resourced sub-Saharan

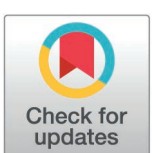

**Data availability statement:** The data underlying this study are available in full in the Supporting Information files submitted with this manuscript. All survey responses used in the analysis have been anonymized and aggregated to protect the identities of participating institutions and individuals. The data is shared as supplemental information as two files – 'CHAMP File quantitative Data' and 'CHAMP Free Text'.

**Funding:** The author(s) disclosed receipt of the following financial support for the research, authorship, and/or publication of this article: This study was funded by the EMLA Foundation (institutional grant to PL; WJ; and LS). There was no number associated with the grant. The funders had no role in study design, data collection and analysis, decision to publish, or preparation of the manuscript.

**Competing interests:** The authors have declared that no competing interests exist.

African hospitals serving children. The aim was to evaluate hospital facilities, infrastructure, equipment, supplies, services, staffing, and readiness to care for children amid public health emergencies. This report analysed a subset of survey questions that characterised the hospitals and assessed facilities, equipment, supplies, infrastructure and capacity to respond to emergencies and outbreaks. Twenty-four sites were recruited. Twenty hospitals from 15 countries completed the survey from 2018 to 2019. This portion of the CHAMP study identified issues with facilities, equipment, supplies, infrastructure, and the capacity to respond to emergencies and infectious disease outbreaks. On a day-to-day basis, most hospitals were operating at or near capacity and frequently experienced power outages and water shortages. Overall, most hospitals were ill-prepared to manage a major disaster or infectious disease outbreak. If countries are to be prepared to deal with current needs as well as to prevent, detect, and rapidly respond to public health threats, hospitals that care for children will require significant investments.

## Introduction

The Joint External Evaluation (JEE) tool was developed by the World Health Organization WHO) [1] to assist countries in assessing their capacities to prevent, detect and rapidly respond to public health threats. However, the tool does not incorporate an evaluation of where and how children are cared for in public health emergencies. Children account for about 25% of the world's population [2] and have special vulnerabilities and needs in emergencies and infectious diseases outbreaks [3,4]. Children's hospitals are often best prepared to meet the medical needs of children during emergencies and generally offer the most complex and comprehensive care for children available in a country [5]. The Children's Hospitals in Africa Mapping Project (CHAMP) was developed to survey sub-Saharan African hospitals serving children, with the aim of evaluating hospital facilities, infrastructure, equipment, supplies, services, staffing and readiness to care for children amid public health emergencies. We have previously reported on CHAMP data detailing the available resources for diagnosis, treatment, and prevention of community acquired infections and the prevention, surveillance, and management of healthcare-associated-infections (HAIs) [6].

## Methods

The study by Tarun [6] provided a description of the study sites, investigators, and the overall design of the multi-country study. Below we expand on the design and development of the survey.

### Survey development

In August 2018, four members of the Columbia University's Department of Pediatrics (the Columbia team) (S1 Table) convened a two-day meeting at Columbia Global Center in Nairobi with 25 individuals from 13 African countries who held hospital, medical school or healthcare-related leadership roles (S1 Table). Those invited to the meeting were known to the Columbia team or were identified through snowball recruitment by those known to the

Columbia team. The healthcare leaders were not compensated or incentivized for their contributions to the survey. The meeting was convened to discuss the development of a survey to assess the readiness of hospitals in sub-Saharan Africa to respond to critical events such as epidemic outbreaks and mass casualties affecting children. While the term "Children's Hospital" may imply a stand-alone facility that exclusively cares for children, during the meeting, the group developed a more practical designation of a CHAMPS "Children's Hospital", as facilities, either hospital building based, or ward based within hospitals, that focus on providing care to neonates, children and adolescents and are capable of providing a broad array of paediatric medical and surgical services, and are staffed by physicians and nurses dedicated to paediatric care with variable degree of formal paediatric training.

Over the two-day meeting, the participants expanded the scope of the survey to include other topics important in assuring children received the highest quality of care possible, e.g., the availability of malnutrition support services. A draft of the survey was shared with the meeting participants in late October for their feedback and subsequently revised and used for the study. The survey included yes/no questions, follow-up questions to further characterise yes responses, multiple choice questions, and free text. We did not seek to guide respondents to specific answers, and no responses were forced. The final survey consisted of 958 questions.

### Site selection and survey implementation

We sought to recruit hospitals that met the CHAMP definition of a "Children's Hospital". Potential eligible sites were suggested by the Nairobi meeting participants, other colleagues with experience working in Africa, and by the study funder, the ELMA Foundation.

Survey responses were entered by each site's investigator or delegate into a password-protected REDCap database managed by investigators at Columbia University Irving Medical Center (CUIMC). Site investigators were encouraged to ask other administrators at their hospital for assistance answering survey questions, e.g., asking the director of surgery about the number of operating theatres available for paediatric surgeries.

### Sub-study survey questions

This sub-study focuses on survey questions relating to hospital characteristics, inpatient and outpatient facilities and normal capacity, surgical and radiology services, equipment and supplies, and infrastructure and capacity to respond to catastrophes and outbreaks. A future paper will deal with paediatric subspecialty services and staffing, laboratory services, and paediatric healthcare training needs.

### Ethics review

Ethics review was conducted at both the local/country level and through CUIMC's institutional review board (IRB) where it was required. Five sites required a formal ethics review. None of the ethics committees approached deemed this study to be human subject research. None of the participating sites needed written informed consent from site investigators and no patients were included or recruited to participate at any site in this study.

### Statistical analysis

The percentage of positive responses for each survey question was calculated and reported. Microsoft Excel software was used for quantitative data analysis. We reported means, frequencies, ranges, and medians with interquartile range (IQR) where relevant.

## Results

### Study sites

Between September 2018 through June 2019, 24 hospitals agreed to participate in the CHAMP study, with 20 hospitals from 15 countries (Cameroon, Ethiopia, Ghana, Kenya, Lesotho, Liberia, Malawi, Nigeria, Rwanda, Sierra Leone, South Africa, Tanzania, Uganda, Zambia, and Zimbabwe). completing the survey between December 2018 and November 2019.

## Hospital characteristics

The characteristics of the hospitals surveyed are presented in Table 1. Sixteen hospitals described their designation as 'national', whilst one as 'regional', another as a paediatric university teaching hospital, one as a tertiary hospital and one as a zonal referral hospital. Seventeen described their institutions as tertiary care facilities, two as secondary care facilities, and one as a primary care hospital. Eight of 20 hospitals were free-standing children's hospitals, and the other 12 were hospitals that had a separate facility or ward for paediatric patients. Nineteen of twenty hospitals were affiliated with a medical school. Sixteen of the hospitals accepted referrals from other hospitals, some of which were more than 50 km away. Nineteen of twenty hospitals surveyed reported the maximum age of children that were allowed for admission in their facilities. The median maximum age allowed for admission was 14 (range 12–18).

## Inpatient facilities

All 20 respondents reported having a separate paediatric in-patient area, and 3 also reported having a separate ward, or facility for adolescents (Table 2). The median number of paediatric beds was 172, and the average daily bed occupancy rate was 87%. The most common causes of high monthly admissions were malaria, respiratory tract infections, diarrheal diseases, meningitis, and malnutrition.

When asked how often the general paediatric wards reached 100% capacity in the past year the most common answers were "almost always", and "most of the time". Fifteen of the 20 hospitals (75%) reported that admitting more than one child in a bed was often necessary. When asked how often that occurred, nine facilities reported daily, one reported weekly, one monthly, three seasonally (e.g., during malaria season), and one reported it occurring rarely. Three hospitals (15%) reported they had an adequate number of beds to meet current needs.

**Table 1. Hospital characteristics.**

| Hospital Characteristics | | % (n/N)[a] |
|---|---|---|
| Designation | National hospital | 94 (16/17) |
| | Regional hospital | 6 (1/17) |
| | District hospital | 0 (0/17) |
| Level of care* | Primary care | 5 (1/20) |
| | Secondary care | 10 (2/20) |
| | Tertiary care | 85 (17/20) |
| Facility type | Public | 80 (16/20) |
| | Private | 5 (1/20) |
| | Public/private partnership | 15 (3/20) |
| Affiliated with Medical School | | 95(19/20) |
| Free Standing Hospital | | 40 (8/20) |
| Separate Ward/Facility for Children | | 100(12/12) |
| Maximum Age of paediatric patients in children's ward (IQR) | | 14 (3) |
| Age admitted to the adult ward, median (IQR) | | 15(1) |
| Accept referrals from other hospitals | | 80(16/20) |
| Number of paediatric patients accepted annually through referrals, median (IQR) | | 6850 (12924) |
| Up-referrals | Do not up-refer | 25 (5/20) |
| | Only within the country | 60 (12/20) |
| | Internationally | 15 (3/20) |

[a] n = positive responses and N = number of hospitals responding to survey questions.

**Table 2. General inpatient wards.**

| General Inpatient Wards | % (n/N)a |
|---|---|
| Has a separate paediatric inpatient area | 100 (20/20) |
| Number of beds, median (IQR) | 172 (171) |
| Average bed occupancy rate, median (IQR) | 87 (27) |
| Sometimes necessary to put more than one child in a bed | 75 (15/20) |
| Has adequate number of beds | 15 (3/20) |
| Number of additional beds needed, median (IQR) | 50 (29) |
| Has protocol for Isolation and cohorting | 35 (7/20) |
| Number of Isolation Rooms, median (IQR) | 2 (5) |
| Has capacity to cohort patients in an infectious disease emergency | 55 (11/20) |
| **Inpatient Surge Capacity** | |
| In a catastrophic event accepts patients above maximum capacity | 60 (12/20) |
| Paediatric inpatient surge capacity elsewhere | 25 (3/12) |

a n = positive responses and N = number of hospitals responding to survey questions.

All 20 hospitals reported a paediatric intensive care unit (PICU) (Table 3 and S2 and S3 Tables). Twelve (60%) reported that they had a separate paediatric unit (PICU) (Table 3), and 12 (60%) reported that their institution had a separate neonatal intensive unit (NICU) (Table 4). Six hospitals had an adult ICU that allocated space for children (S2 Table), and five had a combined NICU/PICU (S3 Table). For the hospitals with a separate PICU (Table 3), the median number of beds was 6 (range, 3–16) The median bed occupancy rate was 83%. Only 1 (8%) of the 12 hospitals reported having an adequate number of PICU beds to meet their needs.

For the hospitals with separate NICUs, the median number of beds was 17.5 (range, 4–50) the average daily census was 26, with an average bed occupancy of 100% (Table 4). The average number of additional beds needed was 20.

In addition to general inpatient wards, PICUs, and NICUs; respondents described other types of inpatient spaces where patients received care, including high-dependency cubicles, high-care bays, step-down units, resuscitation units, and surgical inpatient units. Data on the malnutrition wards can be found in S4 Table.

The most cited areas requiring more beds were general paediatric wards, surgical wards, NICU, PICU, emergency room, and malnutrition, renal, cardiac, oncology, and sickle cell wards.

Barriers to adding more beds included the lack of space, funding and inadequate numbers of healthcare providers to staff the extra beds (S5 Table).

## Outpatient facilities

All but one of 20 hospitals (95%) had a general paediatric outpatient care area separate from the adult area (Table 5). Most patients (80%) were seen within 1–3 hours of presenting but waits could be as long as 8–24 hours. The average number of paediatric outpatients (18 responses) seen on a given day was 119 (range 35–300), and the highest number of paediatric outpatients (14 responses) seen in one day in the past year was 156 (range 40–612). The causes of the high volume included malaria, gastroenteritis, acute respiratory infections, pneumonia, burns, accidents, bronchiolitis and bronchitis, and evaluation of newborns for sepsis and neonatal jaundice.

Eighteen of the 20 hospitals (90%) reported having a separate paediatric emergency area (Table 6), distinct from the adult emergency area. The percent median number of beds in the paediatric emergency areas was 12 (range, 3–60). The average daily number of paediatric patients seen in the emergency area was 33, and the average daily census in the

**Table 3. Paediatric Intensive Care Unit (PICU).**

| Paediatric Intensive Care Unit (PICU) | % (n/N)[a] |
|---|---|
| Number of hospitals with dedicated PICUs | 60 (12/20) |
| Number of Beds in the PICU, median (IQR) | 6 (3.75) |
| Average daily census of paediatric patients, median (IQR) | 5 (1.25) |
| Average Bed Occupancy rate, median (IQR) | 82.5% (21.25) |
| Has an adequate number of beds in the PICU to meet current needs | 8.34 (1/12) |
| Additional beds are needed for paediatric patients, median (IQR) | 5 (8) |
| Isolation rooms in the PICU | 25 (3/12) |
| **PICU Surge Capacity** | |
| Surge Capacity in PICU | 27.3 (3/11) |
| PICU surge capacity elsewhere | 0 (0/8) |
| In an infectious disease emergency has the capacity to cohort paediatric patients in a PICU | 8.3 (1/12) |
| Can cohort paediatric patients in need of intensive care elsewhere | 33 (3/9) |

[a] n = positive responses and N = number of hospitals responding to survey questions.

**Table 4. Neonatal Intensive Care Unit (NICU).**

| Neonatal Intensive Care Unit (NICU) | % (n/N)[a] |
|---|---|
| Number of hospitals with dedicated NICUs | 60 (12/20) |
| Number of beds in the NICU, median (IQR) | 17.5 (34) |
| Average daily census of paediatric patients, median (IQR) | 26 (36.5) |
| Average Bed Occupancy rate, median (1QR) | 100 (5) |
| Has an adequate number of beds in the NICU to meet current needs | 8.3 (1/12) |
| Additional beds are needed for paediatric patients, median (IQR) | 20 (17.5) |
| Isolation rooms in the NICU | 16.7 (2/12) |
| **NICU Surge Capacity** | |
| Surge Capacity in NICU | 16.7 (2/12) |
| NICU surge capacity elsewhere | 0 (0/10) |
| In an infectious disease emergency has the capacity to cohort paediatric patients in a NICU | 25 (3/12) |
| Can cohort paediatric patients in need of intensive care elsewhere | 33 (3/9) |

[a] n = positive responses and N = number of hospitals responding to survey questions.

emergency area in the past year was 60. Seventy percent of the responding facilities had a mechanism to quickly identify and isolate patients with contagious diseases as they entered the emergency area. Fourteen of 20 hospitals (70%) reported that the wait time in the emergency area was less than one hour.

**Table 5. Paediatric outpatient area.**

| Paediatric Outpatient Area | | % (n/N)ᵃ |
|---|---|---|
| Has separate paediatric outpatient area (251) | | 95 (19/20) |
| Average number of paediatric patients seen in the outpatient area on a given day, median (IQR) (255) | | 55 (102.5) |
| The highest daily census of paediatric patients in the outpatient area in the past year, median (IQR) (256) | | 80 (131.75) |
| Wait time to be seen in the Outpatient Department (264) | < 1 hour | 10 (2/20) |
| | 1-3 hours | 80 (16/20) |
| | 4-8 hours | 5 (1/20) |
| | 8-24 hours | 5 (1/20) |
| | > 24 hours | 0 (0/20) |
| **Outpatient Surge Capacity** | | |
| Surge capacity for paediatric patients in the outpatient area (258) | | 50 (10/20) |
| Paediatric outpatient surge capacity elsewhere (260) | | 10 (1/10) |

ᵃ n = positive responses and N = number of hospitals responding to survey questions.

**Table 6. Paediatric emergency room capacity.**

| Paediatric Emergency Room Capacity. | | % (n/N)ᵃ |
|---|---|---|
| Has a separate paediatric emergency (casualty) area | | 90 (18/20) |
| Number of beds are in the paediatric emergency (casualty) area, median (IQR) | | 12 (7.5) |
| Average number of paediatric patients seen in the emergency (casualty) area on a given day, median (IQR) | | 32.5 (66.75) |
| The highest daily census in the emergency (casualty) area in the past year, median (IQR) | | 60 (128) |
| The most common reasons for high usage were malaria, gastroenteritis, and respiratory infections | | |
| Has a mechanism (screen and triage) to quickly identify and isolate patients with contagious diseases as they enter the emergency (casualty) area | | 70 (14/20) |
| Wait time to be seen in the ER | < 1 hour | 70 (14/20) |
| | 1-3 hours | 25 (5/20) |
| | 4-8 hours | 5 (1/20) |
| | 8-24 hours | 0 (0/20) |
| | > 24 hours | 0 (0/20) |
| **Paediatric Emergency Area Surge Capacity** | | |
| Surge capacity for paediatric patients in the emergency area | | 35 (7/20) |
| Paediatric emergency surge capacity elsewhere | | 14.3 (2/14) |

ᵃ n = positive responses and N = number of hospitals responding to survey questions.

## Isolation rooms and capacity to cohort patients and surge

In the general inpatient wards 19 hospitals (95%) reported that the median number of isolation rooms was 2 (Table 2). In an infectious disease emergency, 11 (55%) hospitals had the capacity to cohort patients in their general inpatient units and 7 (35%) hospitals reported a standard operating procedure for cohorting and isolating patients. In the event of a

catastrophe, e.g., Cholera outbreak, chemical poisoning, 12 hospitals (60%) reported they had surge capacity in their general paediatric wards and 3 of 12 hospitals (25%) reported they had surge capacity elsewhere. When asked, "How do you accommodate excess patients?" the responses included, add "more beds", "vacate stable patients", "open up an isolation unit/holding bay not in regular use", "make use of the short stay ward", "have patients share beds", "use the floor", or "use a separate shelter outside the main hospital".

In hospitals with a dedicated PICU, 3 of 12 (25%) reported having Isolation rooms (Table 3). Three of 11 hospitals (27%) reported having surge capacity in the PICU. In an infectious disease emergency 1 hospital (8.3%) reported the capacity to cohort paediatric patients in the PICU.

In the 12 hospitals with dedicated NICUs, 2 (17%) reported having isolation rooms, 3 (25%) had the capacity to cohort patients, 2 (17%) had surge capacity within the NICU and 3 of 9 hospitals (33%) also reported having the capacity to provide intensive care elsewhere (Table 4).

Seven (35%) reported having surge capacity in the emergency area, and two of 14 respondents (14%) described paediatric emergency surge capacity elsewhere within the facility.

Ten of 20 hospitals (50%) had surge capacity for paediatric outpatient care (Table 5).

### Staffing during catastrophic events

If a catastrophic situation exceeded staff capacity to care for children, 19 of 19 hospitals (100%) reported the following overlapping response strategies: pulling additional staff from other services, recalling staff from leave, making do with existing resources, recruiting from other centres, and diverting patients to other hospitals. When asked how many times in the past month the paediatric services have been put on diversion (i.e., closed to new admissions because the facility has reached the maximum number of patients that could be cared for), 15 hospitals answered as follows: never (12), 1–2 days (1), 4 days (1), and several times (1).

### Disaster response program

Ten of 19 hospitals (53%) had a disaster response program, 11 of 19 (58%) had a disaster response plan, and 6 of 19 (32%) conduct disaster simulations and/or drills (S6 Table). Fifteen of 19 hospitals (79%) reported that disaster response coordination was a Ministry level function.

Eleven of 19 hospitals (58%) had a disaster response team, and 17 hospitals (85%) identified their within-facility methods of disaster communication (S6 Table).

Information regarding disaster funding can be found in (S6 Table).

When asked, "Do you have a policy for treating children in the absence of a parent or guardian?" 10 of 18 (56%) answered yes.

### Surgical services

The survey asked three yes or no questions regarding how patients needing surgery were managed: (1) did they perform surgery on children in their facility, (2) did they up-refer to children for surgery to other hospitals in their network, and (3) did they up-refer outside their network (Table 7). Of the 20 hospitals, 17 (85%) reported performing surgery on children in their facility. Seven hospitals (35%) reported up-referring children for surgery to other hospitals within their network, and 4 (20%) up-referred outside their network of hospitals.

There were a median of two paediatric operating theatres available daily, and the median number of additional theatres needed was two. Only four of 20 hospitals (20%) reported having an adequate number of operating theatres to meet current paediatric surgery needs.

**Table 7. Surgical services.**

| Surgical Services | | % (n/N)[a] |
|---|---|---|
| Places where Paediatric Surgery is performed | Performed in your hospital | 85 (17/20) |
| | Up referred to another hospital in your network of hospitals | 35 (7/20) |
| | Up referred to a hospital outside of your network of hospitals | 20 (4/20) |
| Number of dedicated paediatric operating theatres that are available daily, median (IQR) | | 2 (2) |
| Number of general (adult/paediatric) operating theatres that can be used for paediatric surgery, median (IQR) | | 2 (2) |
| Has an adequate number of operating theatres to meet your current needs for paediatric surgery | | 20 (4/20) |
| Number of additional operating theatres that are needed, median (Range) | | 2 (1 –6) |
| Average number of paediatric surgeries performed per week, median (IQR) | | 12 (20.5) |
| Wait time for an elective surgery | < 1 week | 11.1 (2/18) |
| | 1 week - 1 month | 33.3 (6/18) |
| | 1-3 months | 11.1 (2/18) |
| | 3-6 months | 27.8 (5/18) |
| | > 6 months | 11.1 (2/18) |

[a] n = positive responses and N = number of hospitals responding to survey questions.

The average wait time for elective surgery was variable, with the longest wait time of more than 6 months in 2 of the 18 hospitals and the shortest wait time of less than 1 week in 2 of the hospitals. S7 Table lists the types of anaesthetic agents used and the methods for sterilizing reusable equipment and administering anaesthetic agents.

### Radiology services and equipment

All hospitals had X-ray and ultrasound machines, 70% had CT scanners, and 50% had MRI scanners. S8 Table provides more details on radiology services.

### PICU and NICU equipment and maintenance

Of the 12 hospitals that had separate PICU, all had pulse oximetry, 11 (92%) had oxygen cylinders, and 8 (67%) had continuous positive airway pressure (CPAP) machines. Of the 8 hospitals (67%) with mechanical ventilators (66%), the median number of ventilators was 4, with a median number of 3 functional on any given day. (Table 8). Other PICU equipment is listed in Table 8.

Of the 12 hospitals with dedicated NICUs, all had phototherapy lights and CPAP machines, 11 (92%) had pulse oximeters and oxygen cylinders. Of the five hospitals (42%) with mechanical ventilators, the median number of machines was 2, with a median of one functional ventilator on any given day (Table 9). Other NICU equipment is listed in Table 9.

The five hospitals with Combined NICU/PICUs had a similar equipment and supply profile as found in hospitals with separate NICUs and PICUs but data were lacking on number of mechanical ventilators (S9 Table).

Fourteen of 19 (74%) hospitals reported having equipment maintenance programs. Examples of equipment maintained included mechanical ventilators and other ICU equipment (13), anaesthesia equipment (13), infant incubators (12).

**Table 8. PICU Equipment.**

| Available Equipment, Supplies or Procedures | % (n/N)[a] |
|---|---|
| Pulse oximeters | 100 (12/12) |
| Continuous ECG monitors | 50 (6/12) |
| Invasive pressure monitors | 16.7 (2/12) |
| Mechanical ventilators | 66.7 (8/12) |
| CPAP machines | 66.7 (8/12) |
| Oxygen concentrators | 41.7 (5/12) |
| Oxygen cylinders | 91.7 (11/12) |
| Peritoneal dialysis | 58.3 (7/12) |
| Haemodialysis machine | 8.3 (1/12) |
| Cooling devices for induced hypothermia | 0 (0/12) |
| High frequency oscillators (HFVO) | 0 (0/12) |
| EEG monitors | 8.3 (1/12) |
| Ultrasound machines | 58.3 (7/12) |
| Phototherapy lights | 58.3 (7/12) |
| Central line insertion and maintenance | 33.3 (4/12) |
| **Number of Machines** | **Median (IQR)** |
| Number of mechanical ventilators | 4 (1.5) |
| Number of functional mechanical ventilators | 3 (2.5) |
| Number of CPAP machines | 2 (1) |
| Number of functional CPAPs machines | 1.5 (2) |
| Number of Oxygen Concentrators in the ICU | 2 (0) |
| Number of Oxygen Cylinders in the ICU | 4 (2.5) |
| Number of functional Oxygen Concentrators in the ICU | 2(2) |

[a] n = positive responses and N = number of hospitals responding to survey questions.

## Medical supplies

Thirteen of 19 hospitals (68%) reported not having adequate medical supplies to meet current paediatric needs, and 6 of 19 (32%) reported they reused disposable medical supplies. The most common supplies and medications in short supply included endotracheal tubes appropriate for infants and children (63%), sterile surgical gloves (53%), commonly used antibiotics (47%), oxygen (42%), Insulin (32%), antimalarial drugs (26%), syringes and/or needles appropriate for paediatric use, saline (21%) and recommended paediatric vaccines (21%) (S10 Table). Five of 18 hospitals reported that they also experienced supply shortages during periods of extreme heat, most commonly sterile gloves, IV catheters, and IV fluids. Other items mentioned that were frequently in short supply were medical equipment, mechanical ventilators, and monitors. The most reused single-use items were nasal prongs, face masks, nebulizing kits, pulse oximetry probes, endotracheal tubes, suction tips, tympanic membrane temp covers, and insulin syringes.

## Infrastructure

Sixteen of 19 (84%) hospitals reported having blackouts in their power supply with 19 of 19 reporting they have emergency generators. Eleven of 19 hospitals (58%) reported interruptions in their water supply. The frequency of power and water interruptions is presented in Table 10. Only one hospital reported never having experienced blackouts.

**Table 9. NICU Equipment.**

| Available Equipment, Supplies or Procedures | % (n/N)[a] |
|---|---|
| Pulse oximeters | 91.7 (11/12) |
| Continuous ECG monitors | 25 (3/12) |
| Invasive pressure monitors | 0 (0/12) |
| Mechanical ventilators | 41.7 (5/12) |
| CPAP machines | 100 (12/12) |
| Oxygen concentrators | 58.3 (7/12) |
| Oxygen cylinders | 91.7 (11/12) |
| Peritoneal dialysis | 0 (0/12) |
| Haemodialysis machine | 0 (0/12) |
| Cooling devices for induced hypothermia | 8.3 (1/12) |
| High frequency oscillators (HFVO) | 0 (0/12) |
| EEG monitors | 0 (0/12) |
| Ultrasound machines | 33.3 (4/12) |
| Phototherapy lights | 100 (12/12) |
| Central line insertion and maintenance | 33.3 (4/12) |
| **Number of Machines** | **Median (IQR)** |
| Number of mechanical ventilators | 2 (5) |
| Number of functional mechanical ventilators | 1 (5) |
| Number of CPAP machines | 5 (6) |
| Number of functional CPAPs machines | 5 (6) |
| Number of Oxygen Concentrators in the ICU | 3 (2.5) |
| Number of Oxygen Cylinders in the ICU | 3 (2.5) |
| Number of functional Oxygen Concentrators in the ICU | 4 (9) |

[a] n = positive responses and N = number of hospitals responding to survey questions.

Six of 18 hospitals (33%) reported that their institution exclusively used a paper system for medical records, while 14 of 19 hospitals (74%) reported using a hybrid (electronic and paper) system. None exclusively used an electronic system. Of the 14 hospitals using an electronic system, 13 (93%) reported that they could selectively search the system for specific information.

## Discussion

This part of the CHAMP study characterized the 20 participating hospitals and assessed their facilities, infrastructure, equipment, and capacity to respond to catastrophic emergencies and infectious disease outbreaks.

The 20 hospitals included in this study were national, regional, public, tertiary, and referral hospitals 19 of which were affiliated with medical schools. Given their status, these institutions would be among the leading children's hospitals in their country.

The hospitals had substantial general paediatric ward capacity with a median of 172 beds. Specialized units to provide intensive care to neonates and children were available at 85% of CHAMP hospitals. Our findings, regarding the capacity to provide critical care differ from those of the PediPIPES survey of 37 hospitals in 10 West African hospitals [7]. While the percentage of tertiary facilities was similar for the two studies (85% of CHAMP hospitals versus 87% of PediPIPES facilities), only 43% of the PediPIPES facilities had a NICU or a general intensive care unit, suggesting that the lack of paediatric critical care services for infants and children in Africa is likely greater than observed in our study. A dedicated

**Table 10. Infrastructure.**

| Infrastructure | | % (n/N)ᵃ |
|---|---|---|
| **Electricity** | | |
| Number of hospitals that experience blackouts | | 78.9(15/19) |
| Frequency of blackouts | Daily | 13.3 (2/15) |
| | Weekly | 26.7 (4/15) |
| | Monthly | 13.3 (2/15) |
| | Infrequently | 46.7 (7/15) |
| Number of hospitals that have emergency generators | | 100 (19/19) |
| Frequency at which generators are used during a blackout | Always | 84.2 (16/19) |
| | Sometimes | 15.8 (3/19) |
| | Rarely | 0 (0/19) |
| Number of hospitals that experience other fluctuations in power supply, e.g., power drops, surges, blackouts)? | | 68.4 (13/19) |
| **Water Supply** | | |
| Number of hospitals reporting the frequency of interruptions in water supply | | 52.6 (10/19) |
| Frequency of interruptions to the water supply | Daily | 50 (5/10) |
| | Weekly | 20 (2/10) |
| | Monthly | 10 (1/10) |
| | Infrequently | 20 (2/10) |

ᵃ n = positive responses and N = number of hospitals responding to survey questions.

paediatric emergency room was available at 90% of CHAMP hospitals and most had the ability to quickly identify and isolate patients with suspected infectious diseases. Surgical services and radiological services were available at most hospitals. Regarding radiology equipment, ultrasound and x-ray machines were available at all hospitals and CT and MRI scanners at 70% and 50% of hospitals, respectively. In the 12 hospitals with a separate PICU, all had pulse oximetry, and 67% had mechanical ventilators. In the 12 hospitals with separate NICU, all had CPAP machines and phototherapy lights, and 92% had pulse oximeters. Only about 50% of hospitals had mechanical ventilators, like the PediPIPES hospitals with intensive care units.

Most hospitals faced significant challenges in meeting their current clinical demands, often exceeding bed capacity across general paediatric and specialty paediatric wards, intensive care units, and the emergency room. Bed shortages often necessitated putting more than one child in a bed or crib, posing a risk of hospital-acquired infections, a problem recognized in another study [8]. Eighty percent of hospitals did not have enough operating theatres to meet current needs.

Hospitals also had infrastructure challenges with unreliable sources of power and water, findings similar to those of other published studies [7,9]. Most hospitals reported power outages, a problem predicted to worsen with climate change [10]. Over half the hospitals reported interruptions in their water supply. The availability of a reliable water source is a key component of the WHO's special focus on WASH (water, sanitation, hygiene and health) and infection prevention and control [11]. Investments are needed to ensure hospitals have adequate reliable energy and water supplies that are resilient to damage caused by climate change.

Additional challenges that would impede the CHAMP hospitals in addressing a catastrophe or epidemic included limited numbers of isolation rooms, inability to cohort patients, lack of standard operating procedures for isolation and cohorting, and limited capacity to surge.

Forty-one percent of the Sub-Saharan African population is under 15 years of age, some 593 million children in 2024, with a projection of 734 million children by 2043 [12,13]. Our findings and the projection of continued growth in the size

of the paediatric population indicate there is a critical need to invest in facilities, infrastructure, and equipment. The ability to add additional hospital beds is partly limited by an inadequate supply of doctors and nurses. Africa already faces a shortage of healthcare workers [14], and the supply of paediatricians is woefully inadequate. In high-income countries, the median number of paediatricians per 100,000 children is 72, while in Africa, it is 0.5 per 100,000 children [15]. Essential to improving African hospitals that care for children will be the training of many paediatric physicians, surgeons, nurses and other paediatric healthcare workers.

This study had some important limitations. The hospitals that were included in this study were highly selected and not all invited hospitals responded. Four hospitals that agreed to participate did not complete the surveys. We recognized that this group of hospitals were not representative of most African hospitals where children might receive care. Indeed, we expect that most hospitals where children might receive care have greater challenges and fewer resources than the hospitals in our study. Not all hospitals answered every question. There was no verification of the responses. The survey was conducted at one point in time prior to the COVID-19 pandemic. The determination of 'adequate' or 'inadequate' resources was subjective as these were left to the discretion of the site that completed this survey.

## Conclusions

This study determined that the CHAMP hospital facilities, equipment, supplies and infrastructure were inadequate to support routine clinical demands. The hospitals were, by and large, ill-prepared to respond to public health emergencies affecting children. Greater investments in African hospitals that care for children and in the training of paediatric healthcare workers should be a high priority for governments and donors. Not only is it the right thing to do [16], but it is good economics [17–23]. The CHAMP survey tool (S1 Appendix) is available and might be useful in identifying and prioritizing investments. As regards hospital readiness to deal with health emergencies we recommend the adoption of established international standards such as the WHO Hospital Emergency Response Checklist or the Essential Emergency and Critical Care (EECC) framework [24,25].

Finally, investments that strengthen and grow children's hospitals will help ensure that African children are afforded the right recognized in Article 24 of the United Nations Convention on the Rights of the Child, "to the enjoyment of the highest attainable standard of health and to facilities for the treatment of illness and rehabilitation of health" [26].

## Supporting information

**S1 Appendix. Redcap Survey Tool.**
(DOCX)

**S1 Table. Nairobi Meeting Attendees.**
(DOCX)

**S2 Table. Adult ICU Capacity.**
(DOCX)

**S3 Table. Combined NICU/ PICU Capacity.**
(DOCX)

**S4 Table. Malnutrition Ward.**
(DOCX)

**S5 Table. Reasons that prevent adding additional beds.**
(DOCX)

**S6 Table. Disaster Response.**
(DOCX)

**S7 Table. Anaesthetic Agents.**
(DOCX)

**S8 Table. Radiology Services.**
(DOCX)

**S9 Table. Combined NICU/ PICU Equipment and Supplies.**
(DOCX)

**S10 Table. Medical Supplies.**
(DOCX)

**S1 Data. Quantitative Data of the Survey.**
(XLSX)

**S2 Data. Qualitative (Free Text) Data of the Survey.**
(XLSX)

## Acknowledgments

We would like to thank Dr. Murugi Ndirangu and Ms. Pauline Winfred Muthoni of the Columbia Global Center— Nairobi for helping arrange the August 2018 CHAMP organizing conference and for countless other activities and services that were essential in conducting this study. We also wish to thank Noël Manu, the Columbia University Administrative Coordinator at the time, for her exceptional work in coordinating countless efforts critical to the success of the project. We want to acknowledge the hospital staff who helped gathered this information. We also wish to thank the ELMA Foundation for their financial support of this project.

## Author contributions

**Conceptualization:** Lawrence R. Stanberry, Philip LaRussa, Wilmot James.

**Data curation:** Maitry Mahida.

**Formal analysis:** Vinayak Bhardwaj, Lawrence R. Stanberry, Philip LaRussa, Maitry Mahida.

**Funding acquisition:** Lawrence R. Stanberry, Philip LaRussa, Wilmot James.

**Investigation:** Vinayak Bhardwaj, Lawrence R. Stanberry, Philip LaRussa, Wilmot James, Maitry Mahida, Aimable Kanyamuhunga, Atnafu Mekonnen Tekleab, Augustine Omoigberale, Crispen Ngwenya, David Musorewegomo, Dipesalema Joel, Ezekiel Mupere, Fidelis Ewenitie Eki-Udoko, Hannah Bousquet, Heloise Buys, Hilda Angela Mujuru, IkeOluwa Lagunju, Irene Marete, Jethro Zawolo, Jonathan Kaunda Mwansa, Joseph Tawanda Chava, Maima Kawah Baysah, Mildred Anyango Mudany, Nancy Biyeah Yang Ngum, Nellie V.T. Bell, One Bayani, Pauline Samia, Ruth Nduati, Sam Miti, Schyler Zane Grodman, Thembisile Dintle Mosalakatane, Workeabeba Abebe, Ashraf Coovadia.

**Methodology:** Lawrence R. Stanberry, Philip LaRussa, Aimable Kanyamuhunga, Augustine Omoigberale, Crispen Ngwenya, Ezekiel Mupere, Hannah Bousquet, Heloise Buys, Jonathan Kaunda Mwansa, Mildred Anyango Mudany, Nancy Biyeah Yang Ngum, Ruth Nduati, Workeabeba Abebe, Ashraf Coovadia.

**Project administration:** Lawrence R. Stanberry, Philip LaRussa, Ashraf Coovadia.

**Resources:** Lawrence R. Stanberry.

**Supervision:** Lawrence R. Stanberry, Philip LaRussa, Wilmot James.

**Validation:** Lawrence R. Stanberry.

**Visualization:** Maitry Mahida.

**Writing – original draft:** Vinayak Bhardwaj, Lawrence R. Stanberry, Philip LaRussa, Maitry Mahida, Ashraf Coovadia.

**Writing – review & editing:** Vinayak Bhardwaj, Lawrence R. Stanberry, Philip LaRussa, Wilmot James, Maitry Mahida, Aimable Kanyamuhunga, Atnafu Mekonnen Tekleab, Augustine Omoigberale, Crispen Ngwenya, David Musorewegomo, Dipesalema Joel, Ezekiel Mupere, Fidelis Ewenitie Eki-Udoko, Hannah Bousquet, Heloise Buys, Hilda Angela Mujuru, IkeOluwa Lagunju, Irene Marete, Jethro Zawolo, Jonathan Kaunda Mwansa, Joseph Tawanda Chava, Maima Kawah Baysah, Mildred Anyango Mudany, Nancy Biyeah Yang Ngum, Nellie V.T. Bell, One Bayani, Pauline Samia, Ruth Nduati, Sam Miti, Schyler Zane Grodman, Thembisile Dintle Mosalakatane, Workeabeba Abebe, Ashraf Coovadia.

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
