## [Decision Letter · Decision Letter 0]

10 Jul 2025

PGPH-D-25-01272

The Children’s Hospitals in Africa Mapping Project (CHAMP) Survey: Facilities, Equipment, Supplies, Infrastructure, and Capacity to Respond to Emergencies.

Dear Dr. Coovadia,

Thank you for submitting your manuscript to PLOS Global Public Health. After careful consideration, we feel that it has merit but does not fully meet PLOS Global Public Health’s publication criteria as it currently stands. Therefore, we invite you to submit a revised version of the manuscript that addresses the points raised during the review process.

We look forward to receiving your revised manuscript.

Kind regards,

Vinay Nair Kampalath, MD, MSc, DTM&H

Academic Editor

Journal Requirements:

Additional Editor Comments (if provided):

Thanks for this important contribution. The development of paediatric capacity in Africa is important, particularly as it is relevant to healthcare facilities' capabilities to respond to emergencies. This contribution is vitally important, but we invite you to submit a revision that takes into account the reviewers' comments. In particular, I want to highlight Reviewer 2's extensive feedback on how some restructuring of the paper will strengthen the results and conclusions.

In your revision, please highlight how you've addressed both Reviewers 1 and 2's comments.

Reviewers' comments:

Reviewer's Responses to Questions

**Comments to the Author**

1. Does this manuscript meet PLOS Global Public Health’s publication criteria?

Reviewer #1: Yes

Reviewer #2: Yes

2. Has the statistical analysis been performed appropriately and rigorously?

Reviewer #1: Yes

Reviewer #2: No

3. Have the authors made all data underlying the findings in their manuscript fully available (please refer to the Data Availability Statement at the start of the manuscript PDF file)?

Reviewer #1: Yes

Reviewer #2: Yes

4. Is the manuscript presented in an intelligible fashion and written in standard English?

Reviewer #1: Yes

Reviewer #2: Yes

Reviewer #1: The 'survey development' section should be described in more detail: please define 'healthcare leaders' and 'team from Columbia'. How were the 13 African countries selected? Were the healthcare leaders compensated or incentivized for their contributions to the survey? Additionally, how were the surveyed hospitals selected? Purposive sampling, snowball technique...?

The overall goal of the survey seems to conflict itself at times: initially it is described as an assessment of pediatric readiness during mass casualties and epidemic outbreaks. However, later the participants expanded the scope of the survey to include quite broad measures, like "highest quality of care" and "malnutrition support services." If the scope was just overall pediatric readiness, and not specifically emergencies or mass casualties, the title and abstract should be changed to reflect this.

Please give more information about the IRB process, including the five sites where review was required and protocol number. In line 144, define "CIUMC".

Line 184 seems irrelevant to this study's aims of emergency preparedness: "The most common causes of high monthly admissions were malaria, respiratory tract infections, diarrheal diseases, meningitis, and malnutrition." Lines 228-230 is similar, where if you're talking about emergency preparedness, I don't know that high daily volumes from pneumonia is applicable. Table 6 is similar "most common reasons for high ER usage": if you're trying to connect common diagnoses as potential areas for offloading during a surge, you need to explicitly state that. Otherwise this data seems like it belongs in a different publication.

In Table 2, define "SOP."

Line 205 has a typo: " For the hospitals with separate NICUs, the average number of beds was 17. 5 (range, 4-50)

the average daily census was 26, with an average bed occupancy of 100% (Table 4)."

Table 7: I don't see how average wait time to elective surgery pertains to mass casualties and emergency preparedness.

Line 358 has two periods.

Line 438-439: "Not only is it the right thing to do, but it is good economics." This is an important point that needs to be much more clearly emphasized. Give statistics on how much money is saved by investing in quality pediatric care.

In the limitations section, you state that not all hospitals answered every question. Please include response rates by percentages - the questions that are left unanswered also yield important data.

Reviewer #2: The title addresses a relevant and important topic, particularly for pediatric healthcare in sub-Saharan Africa. The following comments are intended to help strengthen the manuscript’s quality, clarity, and scientific rigor.

Major Comments

1. Sample Size and Selection Criteria

The study does not clearly describe how the 20 hospitals were selected by the participants in Nairobi. The note that “Potential eligible sites were suggested by the Nairobi meeting participants” requires further explanation — detailing how, when, and based on what criteria these sites were proposed, and whether this process introduced selection bias.

Moreover, the fact that 95% (19/20) of the surveyed hospitals are affiliated with medical schools suggests that the sample largely consists of teaching hospitals. This should be acknowledged as a limitation, as it affects the representative of the findings for the broader landscape of hospitals in Africa.

2. Clarification of Terminology

The term “Children’s Hospital” may give the impression that Africa has many stand-alone pediatric hospitals, which is not the case in most countries. The authors should try another wording than children hospital as it gives wrong impression that these hospitals were standalone pediatric hospitals even if it is operational defined

3. Use of WHO Standards

The criteria used to classify medical supplies and services as “adequate” or “inadequate” are not clearly defined. It would strengthen the methodology if the authors referenced established international standards — such as the WHO Hospital Emergency Response Checklist or the Essential Emergency and Critical Care (EECC) framework — to guide and standardize the hospital readiness assessment.

4. Results Section

The sentence “The median ‘maximum age’ of children cared for in the facilities surveyed was 14 years (range: 12–18)” (line 176) requires clarification. It is unclear whether this refers to the maximum age allowed for admission, or the median of the maximum ages reported by each hospital. Both the maximum and median values should be clearly defined and reported consistently.

5. Discussion Section

The discussion section largely restates the results without critically interpreting them in relation to the study’s aim — assessing hospitals' readiness to care for children amid public health emergencies.

Additionally:

The data point that “Fourteen out of 20 hospitals (70%) reported that the wait time in the emergency area was less than one hour” should be discussed more meaningfully. This reported wait time is considerably better than expected in the African context, given known challenges with patient load, staff shortages, and emergency system limitations. The authors should explore possible explanations and discuss this finding within the regional context.

Although the aim mentions readiness for emergencies and disasters, the findings are not adequately discussed within the framework of emergency and disaster preparedness standards. The authors should reflect on how the reported infrastructure, equipment, and services compare to established emergency and disaster readiness benchmarks.

6. Conclusion (in both Abstract and Main Text)

Critical findings such as:

The necessity of admitting more than one child in a bed.

The infection risks associated with placing multiple children in one crib

Frequent interruptions in water supply

should be explicitly highlighted, as these have significant implications for hospital readiness, infection control, and patient safety.

7. Study Limitations

The limitations of the study should be explicitly stated, including:

The nature of the hospitals surveyed (mostly tertiary, teaching hospitals)

The sample’s lack of representative for the wider African hospital landscape

Potential selection bias

Editorial Comments

The aim of the study is unnecessarily repeated between lines 91–96 and again in lines 97–98. This duplication should be removed for clarity.

Key words: can be focused to the research

**Do you want your identity to be public for this peer review?** For information about this choice, including consent withdrawal, please see our Privacy Policy

Reviewer #1: No

Reviewer #2: **Yes: ** Tigist Bacha

---

## [Decision Letter · Decision Letter 1]

23 Sep 2025

PGPH-D-25-01272R1

The Children’s Hospitals in Africa Mapping Project (CHAMP) Survey: Facilities, Equipment, Supplies, Infrastructure, and Capacity to Respond to Emergencies.

Dear Dr. Coovadia,

Thank you for submitting your manuscript to PLOS Global Public Health. After careful consideration, we feel that it has merit but does not fully meet PLOS Global Public Health’s publication criteria as it currently stands. Therefore, we invite you to submit a revised version of the manuscript that addresses the points raised during the review process.

We look forward to receiving your revised manuscript.

Kind regards,

Vinay Nair Kampalath, MD, MSc, DTM&H

Academic Editor

Journal Requirements:

Additional Editor Comments (if provided):

Hi authors, there are some very minor revisions that require attention for editing and language. Please address these and resubmit.

Reviewer #1: Just a few typos: in line 74 there should be two different sentences. Twenty-four sites were recruited. Twenty hospitals from 15 countries completed the survey from 2018 to 2019.

Line 86 should not have a comma after 'where'

Line 100 should not have a comma after 'Tarun'

Line 161 should have a space between Sierra and Leone

Reviewer #2: The revised manuscript can be considered for publication, but it still requires English language editing. I am also uncertain about the reference style (e.g., references 17–23); these may be summarized rather than listed individually. This is a minor issue.

Additionally, in line 198 the authors state that all hospitals have a PICU, yet in the discussion they describe a shortage of pediatric intensive care units. This inconsistency should be corrected.

The conclusion could also be improved: some parts may be moved to the discussion, while the conclusion itself should be concise, summarized, and include the study’s limitations.

Thank you.

Reviewers' comments:

Reviewer's Responses to Questions

**Comments to the Author**

Reviewer #1: All comments have been addressed

Reviewer #2: All comments have been addressed

publication criteria?

Reviewer #1: Yes

Reviewer #2: Yes

3. Has the statistical analysis been performed appropriately and rigorously?

Reviewer #1: Yes

Reviewer #2: Yes

4. Have the authors made all data underlying the findings in their manuscript fully available (please refer to the Data Availability Statement at the start of the manuscript PDF file)?

Reviewer #1: Yes

Reviewer #2: Yes

5. Is the manuscript presented in an intelligible fashion and written in standard English?

Reviewer #1: Yes

Reviewer #2: Yes

Reviewer #1: Just a few typos: in line 74 there should be two different sentences. Twenty-four sites were recruited. Twenty hospitals from 15 countries completed the survey from 2018 to 2019.

Line 86 should not have a comma after 'where'

Line 100 should not have a comma after 'Tarun'

Line 161 should have a space between Sierra and Leone

Reviewer #2: (No Response)

**Do you want your identity to be public for this peer review?** For information about this choice, including consent withdrawal, please see our Privacy Policy

Reviewer #1: No

Reviewer #2: **Yes: ** Tigist Bacha

---

## [Editor Report · Decision Letter 2]

20 Oct 2025

The Children’s Hospitals in Africa Mapping Project (CHAMP) Survey: Facilities, Equipment, Supplies, Infrastructure, and Capacity to Respond to Emergencies.

PGPH-D-25-01272R2

Dear Prof. Coovadia,

We are pleased to inform you that your manuscript 'The Children’s Hospitals in Africa Mapping Project (CHAMP) Survey: Facilities, Equipment, Supplies, Infrastructure, and Capacity to Respond to Emergencies.' has been provisionally accepted for publication in PLOS Global Public Health.

Best regards,

Vinay Nair Kampalath, MD, MSc, DTM&H

Academic Editor